# Functional Analysis of a Multiple-Domain CTL15 in the Innate Immunity, Eclosion, and Reproduction of *Tribolium castaneum*

**DOI:** 10.3390/cells12040608

**Published:** 2023-02-13

**Authors:** Suisui Wang, Huayi Ai, Yonglei Zhang, Jingxiu Bi, Han Gao, Peng Chen, Bin Li

**Affiliations:** Jiangsu Key Laboratory for Biodiversity and Biotechnology, College of Life Sciences, Nanjing Normal University, Nanjing 210023, China

**Keywords:** C-type lectin X, *Tribolium castaneum*, RNAi, innate immunity, eclosion, reproduction

## Abstract

C-type lectin X (CTL-X) plays critical roles in immune defense, cell adhesion, and developmental regulation. Here, a transmembrane CTL-X of *Tribolium castaneum*, *TcCTL15,* with multiple domains was characterized. It was highly expressed in the early and late pupae and early adults and was distributed in all examined tissues. In addition, its expression levels were significantly induced after being challenged with pathogen-associated molecular patterns (PAMPs) and bacteria. In vitro, the recombinant TcCTL15 could recognize bacteria through binding PAMPs and exhibit agglutinating activity against a narrow range of bacteria in the presence of Ca^2+^. RNAi-mediated *TcCTL15*-knockdown-larvae infected with *Escherichia coli* and *Staphylococcus aureus* showed less survival, had activated immune signaling pathways, and induced the expression of antimicrobial peptide genes. Moreover, silencing *TcCTL15* caused eclosion defects by impairing ecdysone and crustacean cardioactive peptide receptors (CCAPRs). Suppression of *TcCTL15* in female adults led to defects in ovary development and fecundity, accompanied by concomitant reductions in the mRNA levels of vitellogenin (*TcVg)* and farnesol dehydrogenase (*TcFDH)*. These findings imply that TcCTL15 has extensive functions in developmental regulation and antibacterial immunity. Uncovering the function of TcCTL15 will enrich the understanding of CTL-X in invertebrates. Its multiple biological functions endow the potential to be an attractive target for pest control.

## 1. Introduction

Insects lack an adaptive immune response and rely solely on a powerful innate immunity to resist invading pathogens [1]. When pathogen invasion occurs, an innate immune response is activated by a variety of pattern recognition receptors (PRRs), which can specifically recognize non-self and bind to the evolutionarily conserved pathogen-associated molecular patterns (PAMPs) on the surfaces of pathogens, triggering a series of extracellular and intracellular signal cascade reactions, such as Toll-like receptors, immune deficiency (IMD), and prophenoloxidase pathways, ultimately killing and eliminating invasive pathogens [2]. To date, several types of PRRs corresponding to different PAMPs have been investigated in various insect orders and found to participate in innate immunity, including C-type lectins (CTLs), scavenger receptors, Toll-like receptors, and peptidoglycan-recognition proteins [3,4].

CTLs belong to the lectin superfamily possessing one or more carbohydrate-recognition domains (CRDs), which exist widely in many organisms, ranging from animals and plants to viruses [3,5]. Based on their domain structure and phylogenetic relationships, CTLs can be sorted into three different groups: CTL-S, which possesses only one CRD; IML, with two CRDs; and CTL-X, with one CRD and additional conserved domains. In a short period of evolutionary time, gene duplication and merging produced a diversity of members in the IML group, whereas there were fewer changes in the CTL-S and CTL-X groups, implying that they might have conserved functions [3]. During the past few years, many CTL-Ss in *Tribolium castaneum* Herbst have been identified and proven to participate in the immune response [6,7,8]. To date, a few CTL-Xs were reported to execute immune functions. In *Aedes aegypti*, CLSP2 (CTL-X) consists of a CRD and an elastase-like serine protease and is a negative modulator of hemolymph melanization and antifungal immune responses [9,10]. Moreover, except for the CRD, CTL-X contains additional domains that can carry out other effector functions [11]. In *Drosophila melanogaster* Meigen, furrowed (CTL-X) plays important roles in the development of the eye and bristle [12,13]. Another CTL-X in *D. melanogaster*, contactin, mediates cell adhesion and is required for the organization of septate junctions and paracellular barrier function [14].

CTL15 (CTL-X) is a putative transmembrane protein that shows strong homology to the selectin family in vertebrates [15] and has previously been identified in *Drosophila*, named furrowed [12]. Selectins, members of vertebrate CTLs, are integral transmembrane proteins consisting of a single CRD, an epidermal growth factor like domain, and a variable number of complement control protein (CCP) domains in the extracellular region, a transmembrane domain, and a short cytoplasmic tail in the extracellular region [12,15]. Furthermore, selectins have been found to mediate leukocytes and endothelia adhesion via their CRD in the mammal immune system [16,17]. An intriguing finding was that the second CCP domain of furrowed mediates direct homophilic adhesion and *furrowed* mutants disrupt the development of sensory organs [13]. Selectin family members, as cell adhesion molecules, can be involved in many developmental processes in *Drosophila* and immune responses in vertebrates. The investigation of selectin function in model insects will provide a useful approach to uncover whether invertebrate selectin participates in innate immunity and some biological processes.

The red flour beetle, *T. castaneum,* has become one of the most destructive pests of stored commodities such as cereals and grains in tropical and subtropical areas and causes quantitative losses of stored products [18,19]. Previously, a total of 17 CTLs were characterized in this Coleoptera model species, and 6 (CTL11, CTL12, CTL14, CTL15, CTL16, and CTL17) of the 17 CTLs were grouped into the CTL-X group [20]. Among them, CTL14, CTL15, CTL16, and CTL17 contain a single transmembrane domain, whereas CTL11 and CTL12 lack transmembrane domains. In our recent work, *TcCTL12* was involved in innate immunity, the development of epidermis and muscle, and reproduction [21]. However, transmembrane CTL-Xs have not been well studied in *T. castaneum*. Additionally, it was reported that many membrane proteins play important roles in various biological processes and are considered drug and pesticide targets [22,23]. In this regard, *TcCTL15* was representatively selected to study the multiple potential roles of transmembrane CTL-Xs in *T. castaneum*.

## 2. Materials and Methods

### 2.1. Insects

The Georgia-1 (GA-1) strain of *T. castaneum* was reared in the laboratory as previously described [7] on an artificial diet consisting of wheat flour with 5 % (*v*/*v*) brewer’s yeasts under constant conditions of 30 °C and 40 % relative humidity with a 14 h light/10 h dark photoperiod in a climatic chamber. To collect synchronous eggs, all the female and male adults were placed into jars, allowed to lay eggs, and then removed 12 h later.

### 2.2. Bioinformatic Analysis of TcCTL15

Homology analyses were performed using the BLASTX tool in Genbank from the National Center for Biotechnology Information (NCBI). The MEGA 6.0 software (https://www.megasoftware.net/ accessed on 27 January 2023) was utilized to align multiple protein sequences of TcCTL15 and its homologous CTLs. The alignment was depicted with ESPript 3.0 (https://espript.ibcp.fr/ESPript/cgi-bin/ESPript.cgi accessed on 27 January 2023). The presence and position of the signal peptide were determined using the Signalp5.0 program (https://services.healthtech.dtu.dk/sevice.php?SignalP-5.0 accessed on 9 February 2023). The Simple Modular Architecture Research Tool (SMART) (http://smart.embl-heidelberg.de/ accessed on 27 January 2023) server was applied to predict the motif features of TcCTL15. The isoelectric point (PI) and molecular weight (MW) of TcCTL15 were determined using the ExPASy online tool Compute pI/Mw (https://web.expasy.org/compute_pi/ accessed on 27 January 2023).

### 2.3. RNA Extraction, cDNA Synthesis, and Quantitative Real-Time PCR (qRT-PCR)

To manifest the tissue distribution, samples were collected from the integument, fat bodies, central nervous system (CNS, including the brain, thoracic ganglia, and ventral nerve cord), gut, and hemocytes of 15-day-old larvae. To detect the stage-specific expression, different samples were randomly selected from multiple individuals at various developmental stages: early eggs (EE, 1-day-old eggs), late eggs (LE, 3-day-old eggs), early larvae (EL, 1-day-old larvae), late larvae (LL, 20-day-old larvae), early pupae (EP, 1-day-old pupae), late pupae (LP, 5-day-old pupae), early adults (EA, 1-day-old adults), and late adults (LA, 10-day-old adults). Three biological replicates were performed for each sample.

Total RNAs from whole *T. castaneum* bodies or different tissues were extracted using RNAiso™ Plus (Takara, Dalian, China). The concentration and integrity of RNA were determined by measuring the absorbance at 260 nm/280 nm using a spectrophotometer (Thermo Fisher Scientific, Wilmington, DE, USA) and 1% agarose gel electrophoresis, respectively. HiScript II Reverse Transcriptase (Vazyme, Nanjing, China) was used to reverse-transcribe first-strand cDNA using 500 ng RNA samples.

qRT-PCR was used to detect the expression levels of target genes using SYBR Green Master Mix (Vazyme, Nanjing, China). The primer sequences used for qRT-PCR are provided in Appendix A. qRT-PCR was performed in triplicate for each biological sample using the StepOnePlus Real-Time PCR System (Applied Biosystems, Foster City, CA, USA) based on our previous method [21]. The transcript levels of target genes were normalized against *T. castaneum* ribosomal protein S3 (*RPS3*) and quantified using the 2^−ΔΔCT^ method [24]. All experiments were performed with at least three biological replications.

### 2.4. Immune Challenges

To investigate the changes in *TcCTL15* transcript levels upon immune challenge, each larva was injected with 200 nL of injection buffer (IB, 373 g/L KCl and 0.038 g/L Na_3_PO_4_ 12H_2_O, pH 7.0) containing bacteria (*Staphylococcus aureus* Rosenbach (approximately 7.0 × 10^4^ cells) or *Escherichia coli* Migula (approximately 7.8 × 10^4^ cells)) or PAMPs (lipopolysaccharides (LPS) or peptidoglycan (PGN), 0.08 mg/mL), whereas the larvae were injected individually with 200 nL of IB as the control group. After injection, all larvae were cultured in the same conditions mentioned above. At 12, 24, 36, 48, 60, and 72 h post-challenge, three surviving individuals were randomly selected for RNA extraction and cDNA synthesis. Subsequently, qRT-PCR was employed to determine *TcCTL15* expression profiles. Each sample included three biological replicates.

### 2.5. Recombinant Expression, Purification, and Western Blot Analysis

A 444 bp DNA fragment coding the TcCTL15-CRD domain was amplified using the primers listed in Appendix A. After that, the amplicons were digested with *EcoR* I and *Xho* I and then ligated to the pET-28a expression vector (Novagen, Darmstadt, Germany). The pET-28a-TcCTL15 construct was transformed into *E. coli* strain BL21 (DE3) competent cells (TransGen, Beijing, China) via induction with a final concentration of 0.5 mM isopropyl-β-D-thiogalactopyranoside (IPTG) at 37 °C for 4 h. The recombinant TcCTL15 (rTcCTL15) was expressed as inclusion bodies and subjected to denaturation, refolding, and purification according to a method described earlier [7]. The purified rTcCTL15 fused with a His-tag was checked using 12.5% sodium dodecyl sulfate–polyacrylamide gel electrophoresis (SDS-PAGE) and Western blotting with anti-His mouse polyclonal antibodies (TransGen, Beijing, China). The successfully purified rTcCTL15 was quantified using the total protein quantitative assay kit (Nanjing Jiancheng Bioengineering Institute, Nanjing, China) with bovine serum albumin (BSA) as a standard and was then ready for the binding and agglutination assays.

### 2.6. Microorganism Binding and Agglutination Assays

Gram-positive bacteria (G^+^) (*S. aureus*, *Bacillus subtilis* Ehrenberg, and *Bacillus thuringiensis* Berliner) and Gram-negative bacteria (G^−^) (*E. coli* and *Pseudomonas aeruginosa* Schroeter) were used for the binding assay. Microbes in the mid-logarithmic phase were collected via centrifugation at 12,000× *g* for 1 min, washed three times with sterilized Tris-buffered saline (TBS), and then resuspended in TBS. The purified rTcCTL15 (600 μg/mL) was mixed with a microbe suspension in the presence or absence of 10 mM CaCl_2_ with gentle rotation at 28 °C for 2 h. Then, the bacteria were pelleted via centrifugation and washed with TBS five times. The microorganism pellets were eluted with a 7% SDS solution for Western blot analysis.

As mentioned above, five microorganisms were also used to evaluate the agglutination activity of rTcCTL15. An agglutination assay was performed as previously reported [25]. Bacteria cultured in an LB medium overnight were harvested via centrifugation, washed three times with TBS, resuspended, and diluted in TBS at 2.0 × 10^8^ cells/mL. Subsequently, the purified rTcCTL15 (200 μg/mL) and recombinant methyl-CpG-binding domain (rTcMBD) in *T. castaneum* (as a negative control) were incubated with a microorganism suspension in the presence or absence of 10 mM CaCl_2_ for 1 h at 28 °C. The agglutination reactions were observed and photographed under a fluorescent microscope (Nikon, Shanghai, China).

### 2.7. Polysaccharide-Binding Assay

The binding activity of rTcCTL15 to PAMPs was examined using an enzyme-linked immunosorbent assay (ELISA) according to previous reports [8]. Briefly, 50 μL of LPS or PGN (80 μg/mL) was coated on the wells of a 96-well microplate at 37 °C overnight and then evaporated at 60 °C for 30 min. After blocking with BSA (100 μL/well of 1 mg/mL) in TBS for 2 h at 37 °C, 100 μL serial concentrations of rTcCTL15 (0–70 μg/mL) were added to each well and incubated at room temperature for 3 h. The plate was washed with TBS, and 100 μL/well of an anti-His mouse polyclonal antibody (TransGen, Beijing, China) diluted to 1:2000 with TBS was added and incubated at 37 °C for 1 h. The plate was rewashed with TBS, and 100 μL/well of a goat anti-mouse IgG horseradish peroxidase (HRP) conjugate (TransGen, Beijing, China) diluted to 1:5000 with TBS was added and incubated at 37 °C for 1 h. After washing with TBS, 100 μL/well of a soluble 3,3′,5,5′-tetramethylbenzidine (TMB) solution (Solarbio, Beijing, China) was added and incubated at 28 °C for 10 min under darkness and then stopped with 100 μL/well of 2 mol L^−1^ H_2_SO_4_. After the termination of the reaction, an automatic enzyme-linked immunosorbent assay reader (BioTek, Winosky, VT, USA) was utilized to measure the absorbance at 450 nm. Each binding assay was repeated three times.

### 2.8. Double-Stranded RNA Synthesis and RNAi Assay

To synthesize double-stranded RNA (dsRNA), DNA templates of *TcCTL15* and *TcVer* were amplified using a T7 promoter sequence at the 5′-end-linked primers. The PCR products (*TcCTL15* (490 bp) and *TcVer* (609 bp)) were purified and were utilized to synthesize dsRNA in vitro using the TranscriptAid^TM^ T7 High Yield Transcription Kit (Thermo Fisher Scientific, Shanghai, China). After synthesis, agarose gel electrophoresis was used to verify the quality and size of the dsRNA. Each dsRNA was quantified with spectrophotometry and diluted with diethylpyrocarbonate-treated water at an equal concentration of 2 μg/μL. The same amount of ds*TcVer* for the specific visible phenotype in the eyes was used as a control [26].

For the dsRNA-dependent knockdown experiment, a total of 200 ng of the ds*TcCTL15* or ds*TcVer* was injected into the hemocoels of each larva using an InjectMan 4 (Eppendorf, Hamburg, Germany). On day 3 post-injection, three individuals were randomly collected from the different experimental groups for RNA extraction and cDNA synthesis. The efficiency of dsRNA-mediated gene silencing was further confirmed using qRT-PCR. To study the effect of *TcCTL15* on reproduction, dsRNA was injected into 5-day-old female and male pupae. Each microinjected pupa that developed to the adult stage was individually matched with a pupa with a different sex and then transferred into a glass bottle. The total number of eggs was counted and calculated for each adult pair for three consecutive days after mating. Meanwhile, to explore the roles of *TcCTL15* in growth and development, ds*TcCTL15* or ds*TcVer* were injected into the 12-day-old larvae and 1-day-old pupae. In addition, to evaluate the larval survival rate after *TcCTL15* knockdown followed by bacterial infection, the *TcCTL15*-deficient or *TcVer*-deficient larvae were injected with *E. coli* or *S. aureus*. The number of dead individuals was counted every 24 h for the next seven days.

### 2.9. Expression Profiling of Transcription Factors and AMP Genes

To investigate the potential function of *TcCTL15* in regulating the expression of downstream immune-related genes, the ds*TcCTL15*-injected larvae were challenged with *E. coli* or *S. aureus*. In the RNAi following bacterial injection, qRT-PCR was used to determine the mRNA levels of nine AMP genes and four transcription factor genes. The primers used for the qRT-PCR analysis of these immune genes are shown in Appendix A.

### 2.10. Statistical Analysis

All analyses were carried out using GraphPad Prism v.8.3 (GraphPad Software, Inc., San Diego, CA, USA), and data are expressed as means ± SDs of triplicate experiments. The statistical significance between treatments was assessed using a one-way ANOVA followed by Tukey’s honestly significant difference (HSD) test. Differences between the means of treatment and control groups were compared using Student’s *t*-test for unpaired samples. Statistical significance was set at *p* ≤ 0.05.

## 3. Results

### 3.1. Molecular Characteristics and Bioinformatics Analysis of TcCTL15

In the *T. castaneum* genome database, the gene encoding TcCTL15 was annotated as TC030754 and mapped to chromosome 3 (LG3). The 3336-base-pair open reading frame (ORF) of *TcCTL15* encodes a single transmembrane protein with 1111 amino acid residues, a predicted molecular mass of 122 kDa, and an isoelectric point (pI) of 5.36. The protein domain analysis indicated that TcCTL15 had one coagulation factor 5/8 C-terminal domain, eleven CCP domains, one canonical CRD with a theoretical pI of 7.77, and a transmembrane region, whereas the signal peptide was absent. In addition, a WND (Trp-Asn-Asp) motif existed in the CRD of TcCTL15 (Appendix A).

The alignment of *TcCTL15* shared 80.4%, 71.0%, 66.6%, 64.4%, 59.6%, and 54.5% identities with those of Aglsushi (*Anoplophora glabripennis*, XP_018565850.1), Znsushi (*Zootermopsis nevadensis*, XP_021932272.1), Nlsushi (*Nilaparvata lugens*, XP_039286815.1), Amsushi (*Apis mellifera*, XP_006559048.2), Bmsushi (*Bombyx mori*, XP_004932837.1), and Dmfurrowed (*D. melanogaster*, NP_001285146.1), respectively. An amino acid sequence alignment of *TcCTL15* with homologous *CTLs* from other species indicated that they all have a strictly conserved CRD domain, a WND motif, and six highly conserved cysteine residues that may form disulfide bonds and may be essential for the stability of structure folding and function in their CRD (Appendix A). These results suggest that *CTL15* evolved conservatively in different insects.

### 3.2. Spatiotemporal Profiles and Induced Expression of TcCTL15

To better investigate the characteristics of TcCTL15, its expression patterns were analyzed in different stages and larval tissues using qRT-PCR. The results revealed that *TcCTL15* was expressed at all examined developmental stages and was especially highly expressed in the early pupae, late pupae, and early adults (Figure 1A). Moreover, *TcCTL15* was expressed in multiple tested tissues of 15-day-old larvae, with the greatest abundance being localized to the integument, followed by the CNS, fat bodies, and hemocytes (Figure 1B). Figure 2 shows the time-course expression profiles of *TcCTL15* after injections of PAMPs or bacteria. The *TcCTL15* mRNA level was significantly up-regulated at 12 and 24 h and at 72 h post-infection with LPS and PGN, respectively (Figure 2A,B). The mRNA abundance of *TcCTL15* was significantly increased at 60 and 24 h after infection with *E. coil* and *S. aureus*, respectively (Figure 2C,D). To analyze the immune role of *TcCTL15* in vivo, we knocked down the *TcCTL15* expression using RNAi following bacterial infection and assessed the survival rate relative to the control. *TcCTL15*-deficient larvae infected with *E. coli* or *S. aureus* showed less survival than *TcVer*-deficient larvae infected with *E. coli* or *S. aureus* (Figure 2E). These results indicate that *TcCTL15* modulates immune activation in *T. castaneum*.

### 3.3. rTcCTL15 Binds to and Agglutinates Bacteria

To further elucidate the role of TcCTL15 in antibacterial immunity, the recombinant plasmid pET-28a-TcCTL15 was expressed using an *E. coli* expression system. After IPTG induction, rTcCTL15 was successfully expressed as inclusion bodies (Figure 3A). Then, rTcCTL15 was purified using denaturation and renaturation and Ni^2+^ affinity chromatography. The SDS-PAGE and Western blotting results confirmed that purified rTcCTL15 migrated as a single distinct band (Figure 3A,B), showing that rTcCTL15 was successfully purified. Then, the potential binding to various bacteria of rTcCTL15 was analyzed. The Western blotting results revealed that rTcCTL15 could directly bind to all the tested bacteria in the presence of CaCl_2_, while no binding activity was detected in the absence of CaCl_2_ (Figure 4A). To clarify whether the microbial binding ability of rTcCTL15 was mediated by cell surface PAMPs, we then examined the potential binding of rTcCTL15 to two typical PAMPs (LPS and PGN) using an ELISA assay. The results showed that rTcCTL15 could bind to these ligands in a dose-dependent manner (Figure 4B). These results suggest that TcCTL15 is capable of binding to various bacteria by recognizing and interacting with their cell surface PAMPs.

The agglutination activity of rTcCTL15 was detected via incubation with five microorganisms. In the presence of Ca^2+^, rTcCTL15 agglutinated the G^+^ *S. aureus* and *B. thuringiensis* and the G^-^ *E. coli* and *P. aeruginosa* but not the G^-^ *B. subtilis* (Figure 4C). Notably, the agglutinating activity was completely lost in the absence of Ca^2+^ and in the rTcMBD group (Figure 4C), which demonstrated that TcCTL15 is a Ca^2+^-dependent C-type lectin. Based on these results, we concluded that TcCTL15 protects *T. castaneum* against pathogens via agglutinating pathogens.

### 3.4. Involvement of TcCTL15 in Regulating the Expression of Transcription Factors and AMP Genes

To address whether *TcCTL15* regulates the expression of transcription factors and AMP genes during bacterial infection, *S. aureus* or *E. coli* infections were performed at 3 d after ds*TcCTL15* or ds*TcVer* injections. The 12-day-old larvae were, respectively, injected with equal volumes of ds*TcCTL15* or ds*TcVer*. After 3 d, 15-day-old larvae were infected with *E. coli* or *S. aureus*, respectively. The mRNA abundances of transcription factors and AMP genes were assessed using qRT-PCR at 24 h after the injection of *E. coli* or *S. aureus*. The ds*TcVer* + *E. coli* and ds*TcVer* + *S. aureus* groups were used as control groups. The results revealed that *TcCTL15* was dramatically down-regulated post-dsRNA-treatment following an *E. coli* or *S. aureus* challenge, and the RNAi efficiencies (4 d after dsRNA injection) were 71.4% and 89.5%, respectively (Figure 5A,D), compared with those of ds*TcVer* injections. Moreover, compared with the control beetles (ds*TcVer + E. coli*), transcription factors (*Rel* and *Stat*) and AMPs, including *attacin2* (*att2*), *attacin3* (*att3*), *defensin3* (*def3*), *cecropin2* (*cecr2*), and *cecropin3* (*cecr3*), were markedly down-regulated in the *TcCTL15*-silenced larvae challenged with *E. coli* (Figure 5B,C). Meanwhile, the expression levels of transcription factors (*Dif1*, *Rel,* and *Stat*) and AMPs, including *attacin1* (*att1*), *att2*, *defensin1* (*def1*), *defensin2* (*def2*), and *def3,* were significantly decreased in the *TcCTL15*-silenced larvae challenged with *S. aureus* (Figure 5E,F). All the above results indicate that *TcCTL15* could be involved in the beetle’s antibacterial immune responses by coordinating the gene expression of transcription factors and AMPs.

### 3.5. TcCTL15 Modulates Eclosion

To determine the potential functions of *TcCTL15*, we injected ds*TcCTL15* or ds*TcVer* into the hemocoel of 1-day-old pupae. The RNAi efficiency of *TcCTL15* was confirmed using qRT-PCR, and *TcCTL15* transcripts were down-regulated to approximately 28% after 72 h of ds*TcCTL15* relative to the expression after ds*TcVer* injection (Figure 6B), showing that the knockdown of *TcCTL15* was successful. Visible eclosion defects were observed after *TcCTL15* knockdown (Figure 6A). *TcCTL15*-deficient pupae had reduced viability, with 46% pupa eclosing deficiency and death in comparison to the ds*TcVer* group (Figure 6C).

It is well-known that insect metamorphosis and development are mainly controlled by neuropeptides, ecdysone (20E), and juvenile hormone (JH) [27]. To investigate the mechanisms of suppressed metamorphosis, we detected the mRNA levels of genes involved in the synthesis and metabolism of 20E and JH and the regulation of eclosion after *TcCTL15* silencing. The results showed that the expression of *spook*, *shadow,* and *shade,* which are involved in the 20E synthesis pathway, was down-regulated in *TcCTL15*-deficient pupae, while there was no significant effect on the expression levels of *E74*, *Broad complex* (*Br-C*), and the fushi tarazu factor-1 (*Ftz-F1*) (Figure 7A). The expression level of the key enzyme gene in the JH synthesis pathway, *JH acid methyltransferase* (*JHAMT*), exhibited no obvious changes between *TcCTL15*-deficient pupae and *TcVer*-deficient pupae (Figure 7B). The receptors of eclosion-related neuropeptides, crustacean cardioactive peptide receptors (*CCAPR1* and *CCAPR2*), were significantly decreased after *TcCTL15* knockdown, whereas *Rickets,* encoding the bursicon receptor, did not change (Figure 7B). These results indicate that the eclosion failure caused by *TcCTL15* knockdown is mediated by ecdysone synthesis and eclosion-related regulatory genes. Altogether, these results indicate that *TcCTL15* is necessary for the viability of *T. castaneum* and is involved in eclosion.

### 3.6. Knocking down TcCTL15 Affects Female Egg-Laying and Ovary and Testis Development

To determine the function of *TcCTL15* in *T. castaneum* reproduction, RNAi was used to knock down the expression of *TcCTL15* in 5-day-old pupae. When compared with the wildtype (WT), the silencing of *TcCTL15* resulted in a dramatic reduction in egg numbers (Figure 8B). ds*TcCTL15* beetles laid an average of about 4.76 ± 0.57 eggs/female during 3 d of mating, while about 37.17 ± 2.18 eggs were produced by the WT females. When the mating experiments were performed on WT females and ds*TcCTL15* males, the females laid a normal number of eggs. In contrast, ds*TcCTL15* females crossed with WT males laid fewer eggs (Figure 8B). Further observation of the ovary and testis development status revealed that ovaries were much smaller in the ds*TcCTL15-*injected females than in ds*TcVer-*injected females of the same age (Figure 8D), whereas the testis morphology group exhibited no obvious differences between the ds*TcCL15* and ds*TcVer*-injected groups (Figure 8E).

To investigate the mechanisms of *TcCTL15* in reproduction, we examined the expression of genes involved in the reproduction-related gene *TcVg*, JH-biosynthesis-related genes such as *TcJHAMT* and *TcFDH*, and the transcription factor *TcFOXO*, which has been reported to act in the regulation of Vg expression and JH signaling [28]. As shown in Figure 8C, the expression of *TcVg* and *TcFDH* was significantly down-regulated in ds*TcCTL15*-injected females, whereas the levels of *TcJHAMT* and *TcFOXO* transcripts exhibited no obvious changes between the ds*TcCTL15* and ds*TcVer* groups. Taken together, these results indicate that the knockdown of *TcCTL15* resulted in a reduction in *TcVg* and *TcFDH* at the mRNA level, caused female reproductive capacity and ovary development defects, and finally led to impaired fecundity.

## 4. Discussion

In this study, *TcCTL15* was demonstrated to be a member of the CTL-X group and the sole selectin in the *T. castaneum* genome, which was homologous to that of *D. melanogaster furrowed*. TcCTL15 contained multiple domains, such as CCP, CRD, and a single transmembrane region, which were common to the structure of selectin family members [12,15]. *TcCTL15,* as a PRR, could recognize, bind to, and agglutinate bacteria using its CRD and activate immune signaling pathways to induce the production of AMPs, thus modulating insect innate immunity, which was consistent with the functions of *TcCTL12* [21], indicating that *TcCTL15* could also participate in various immune responses. In addition, CTL-Xs were reported to be involved in developmental regulation in insects [5]. A recent study showed that *TcCTL12* was required for the metamorphosis, reproduction, and muscle remodeling of *T. castaneum* [21]. Here, *TcCTL15* could regulate essential physiological processes, including metamorphosis and reproduction, implying that *TcCTL15* could also be involved in numerous developmental processes. Above all, the major functions of *TcCTL15* in innate immunity and development in *T. castaneum* were in accordance with vertebrate selectins involved in immunity and furrowed (required for growth and development in *D. melanogaster*), which might be a consequence of analogous functional evolution.

It is known that CTLs bind to carbohydrates on the surfaces of microbes in a Ca^2+^-dependent manner through characteristic motifs located in the CRD, such as EPN (Glu–Pro–Asn), QPD (Gln–Pro–Asp), and WND [29,30]. EPN and QPD are specific for mannose and galactose, respectively, and WND can increase the affinity and specificity of CTLs binding to different carbohydrates [29,30]. The carbohydrate-binding region of the TcCTL15 lectin domain (QPN) was a hybrid between QPD and EPN, suggesting that TcCTL15 might recognize both galactose and mannose and bind to more potential carbohydrate ligands. Moreover, the TcCTL15 lectin domain also contained the WND motif, indicating that the multi-motif of TcCTL15 might possess more sites involved in PAMP recognition and binding against various pathogens in *T. castaneum*. In this study, rTcCTL15 was able to bind microbial cell wall components (LPS and PGN) and exhibited strong binding activities to all the tested microbes in the presence of Ca^2+^, suggesting that TcCTL15 could serve as a PRR for the recognition and binding of various foreign pathogens.

In invertebrates, agglutination activity is a principal feature of lectin and a crucial innate immune method of attacking various invading pathogens [31,32]. In the case of rTcCTL15, it displayed agglutinating activity against G^+^ *S. aureus* and *B. thuringiensis* and G^−^ *E. coli* and *P. aeruginosa* but not against G^−^ *B. subtilis* (Figure 4C). As stated above, rTcCTL15 could bind to various microbes and different PAMPs, but it did not possess a wide range of agglutination against all tested microorganisms, suggesting that TcCTL15 was probably a selective CTL with a relatively narrow range of agglutinating activity. It is highly possible that the mechanisms of binding and agglutinating are different and that the binding was the initial step of immunity, while agglutination might rely on other factors [33,34]. A similar microbial agglutination activity has been reported in CsCTL1 from the tongue sole *Cynoglossus semilaevis* and CaNTC from the crucian carp *Carassius auratus* [35,36]. It was suggested that TcCTL15 could perform an antimicrobial activity by agglutinating bacteria.

It has been reported that most CTLs are tissue-specific. Hemocytes are the major tissue of hemolymph lectins and express a variety of immune-related genes to improve immune responses [37,38,39,40,41]. *TcCTL15* was expressed in many immune-related tissues and was mainly detectable in the integument. Our results were consistent with those from *TcCTL6* of *T. castaneum* and *BmCTL-S2* of *Bombyx mori* [7,42]. As an important immune tissue of insects, the integument plays a key role in the immune system by supporting the epithelial barrier against the invasion of microorganisms [43]. The tissue expression profile of *TcCTL15* indicated that it might participate in various immune responses. Moreover, *TcCTL15* transcripts could be significantly up-regulated after LPS, PGN, *E. coli,* and *S. aureus* challenges (Figure 2), showing that *TcCTL15* could be effectively induced by PAMPs and pathogen stimulation and that it might be involved in defending against pathogens. Similar results were also reported in other CTLs of *T. castaneum* [7,8,21,25]. In addition, the expression of *TcCTL15* could also be down-regulated by PAMPs and bacterial treatments (Figure 2), which might indicate that bacteria might resist host immune attacks via decreasing expression of *TcCTL15* [44]. These data suggest that the expression pattern of *TcCTL15* was not fixed and was complex against different immune challenges, thus demonstrating that it also exerted distinct immune functions.

Most studies of CTLs have focused on their roles in regulating AMPs, which are a large group of short peptides that directly kill or clear infected bacteria, fungi, and viruses [45,46]. For instance, *EsLecH* in the crab *Eriocheir sinensis* could regulate AMP expression via JNK phosphorylation in response to bacterial invasion [46]. Similarly, a novel chimeric CTL of the shrimp *Marsupenaeus japonicus* could induce the expression of selected AMPs via the JAK/STAT pathway [11]. In the present study, the *E. coli* challenge could decrease transcription factor (*Rel* and *Stat*) and AMP (*att2*, *att3*, *def3*, *cecr2,* and *cecr3*) expression in *TcCTL15*-silenced larvae. Meanwhile, the *S. aureus* challenge decreased the expression of transcription factors (*Dif1*, *Rel,* and *Stat*) and AMPs (*att1*, *att2*, *def1*, *def2*, and *def3*). Knockdown of *TcCTL15* followed by bacterial infection resulted in a decline in transcription factors and AMP transcripts, which might explain why injections of *E. coli* or *S. aureus* into *TcCTL15*-silenced larvae caused their mortality to increase significantly (Figure 2E). Our data demonstrated that *TcCTL15* might show antimicrobial activities by positively activating classical immune pathways to selectively induce the expression of AMPs in response to *E. coli* and *S. aureus* challenges.

To some extent, the developmental profiles of the genes reflect their functional importance. For example, *TcCTL12* was highly expressed in the late pupae and early adults, implying that it was crucial for the development of epidermis, muscle, and reproduction [21]. Moreover, the high mRNA level of *TcCTL15* was detected in the early and late pupae and early adults (Figure 1A), indicating that it might be involved in the metamorphosis and development of *T. castaneum*. RNAi of *TcCTL15* in 1- and 5-day-old pupae caused suppressed metamorphosis and reproduction, respectively. It is known that insect CTL-Xs are mainly associated with cell adhesion and developmental regulation [5]. However, how CTL-Xs are involved in developmental regulation still needs to be explored.

Insect metamorphosis is mainly controlled by 20E and JH [47]. The ecdysone synthesis genes *spook*, *shadow,* and *shade* were significantly decreased in ds*TcCTL15*-injected beetles. A reduction in the expression of these genes will hinder the synthesis of ecdysone, which will have a fatal impact on the growth and development of insects [48]. The primary response genes of 20E, such as *E74*, *Br-C,* and *Ftz-F1*, have important regulatory roles in insect metamorphosis and development [49,50,51]. However, their mRNA abundance was not significantly affected between the ds*TcCTL15*-injected beetles and ds*TcVer*-injected beetles (Figure 7A). Hence, we demonstrated that the death from eclosion deficiency in *TcCTL15*-silenced pupae might have been largely due to the suppressed ecdysone synthesis. Insect metamorphosis processes are not only affected by 20E and JH but are also regulated by crustacean cardioactive peptide, bursicon, and their receptors [52]. A knockdown of *CCAPR2* and *Rickets* showed eclosion arrest in *T. castaneum* [52]. The phenomenon of eclosion defect was also observed in ds*TcCTL15*-injected individuals. Additionally, *CCAPR1* and *CCAPR2* were down-regulated, which indicates that the knockdown of *TcCTL15* affected the expression of genes related to eclosion regulation, leading to eclosion failure and then mortality.

The involvement of *TcCTL12* (CTL-X) in the reproduction of *T. castaneum* has been preliminarily studied recently [21]. In this study, a significant reduction in the fertility of offspring and dysplastic ovary were observed in ds*TcCTL15*-injected female individuals. To our knowledge, this was the first time that it was found that CTLs can affect ovary development in insects, which differs from the previous findings that the knockdown of *TcCTL12* resulted in thinner oviducts [21]. It is rare for direct evidence to clarify the relationships between CTLs and the reproduction of insects. In the cotton bollworm *Helicoverpa armigera*, CTL14 interacted with Vg and participated in innate immune responses under a *Beauveria bassiana* challenge [53]. The reproductive success of insects mainly depends on Vg biosynthesis and its uptake process during vitellogenesis [54]. The mRNA level of *TcVg* was significantly decreased in ds*TcCTL15*-injected females, suggesting that the reduction in *Vg* levels might be involved in impaired ovary maturation and fecundity. Similarly, the association of the suppression of *CTL1* and decreased *Vg* expression in embryos was also found in *T. castaneum* [6]. However, the detailed mechanisms remain largely unknown. In adults, JH has well-known roles in regulating insect reproduction, stimulating vitellogenesis, and oocyte maturation, and a knockdown of JH biosynthesis or action-related genes affects Vg biosynthesis or uptake [55,56]. Here, we found that an RNAi-mediated knockdown of *TcCTL15* expression significantly down-regulated the mRNA levels of the JH-biosynthesis-related gene *TcFDH*, which might cause a reduction in *Vg* mRNA levels and abnormality of reproduction. Taken together, these results indicate that the knockdown of *TcCTL15* resulted in reductions in *TcVg* and *TcFDH* at the mRNA level, caused female reproductive capacity and ovary development defects, and finally led to impaired fecundity. A recent study found that *TcFOXO* could bind to the conserved sequence motif of *TcFDH* and negatively regulate the expression of *TcFDH* to participate in beetle reproduction [55]. In this study, no difference in *TcFOXO* at the transcript level was observed between ds*TcCL15*-injected group and the ds*TcVer*-injected group. Hence, the underlying mechanisms of *TcCTL15* affecting reproduction need further investigation.

## 5. Conclusions

TcCTL15, a multiple-domain transmembrane protein of CTL-X was identified in *T. castaneum*. rTcCTL15 could directly bind to G^+^ and G^-^ bacteria through certain PAMPs and agglutinate various bacteria in vitro. *TcCTL15* likely mediated the expression of AMPs by regulating classical immune pathways against pathogens in vivo. Moreover, it was required for metamorphosis and reproduction. These results showed that *TcCTL15* plays crucial roles in the innate immunity, metamorphosis, and reproduction of *T. castaneum*, suggesting it might be a potential target transmembrane protein for pest control. Taken together, the current study deepens our understanding of the functional diversification of CTL-X in *T. castaneum* and its underlying mechanisms in regulating immunity, growth, and development.

## Figures and Tables

**Figure 1 cells-12-00608-f001:**
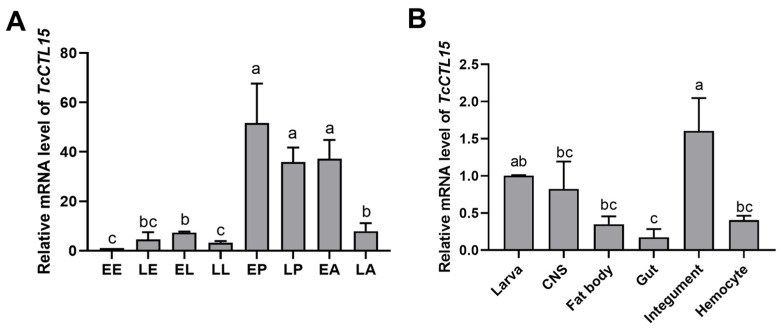
The expression profiles of *TcCTL15* were determined using qRT-PCR in eight stages and different tissues from *T. castaneum*. (**A**) The expression pattern of *TcCTL15* in eight developmental stages. (**B**) The tissue distribution of *TcCTL15* in CNS, fat bodies, gut, integument, and hemocytes from 15-day-old larvae. Transcript levels were normalized using *RPS3* and are shown as means ± SDs from three biological replicates. The different lowercase letters above each bar represent statistically significant differences at the 0.05 level and were analyzed using a one-way ANOVA followed by Tukey’s test.

**Figure 2 cells-12-00608-f002:**
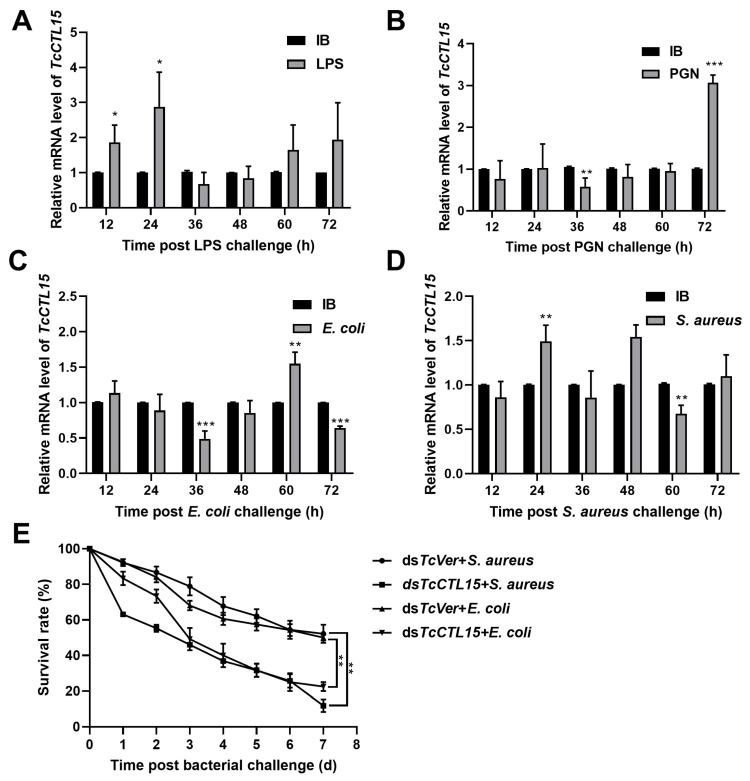
Expression profile analysis of TcCTL15 under immune challenges. (**A**,**B**) qRT-PCR was used to detect the time-course of *TcCTL15* expression after injections of LPS (**A**) and PGN (**B**). (**C**,**D**) qRT-PCR was used to detect the time-course of expression of *TcCTL15* after injections of *E. coil* (**C**) and *S. aureus* (**D**). Fifteen-day-old larvae were infected with LPS (**A**), PGN (**B**), *E. coli* (**C**), *S. aureus* (**D**), or sterile IB (as a control). (**E**) The survival rates of *TcCTL15*-deficient larvae infected with *E. coil* and *S. aureus*. *TcVer*-deficient larvae infected with *E. coil* and *S. aureus* were used as a control. Data were normalized against *RPS3* and are shown as means ± SDs. The asterisks above the bars indicate significant differences. Differences were calculated using Student’s *t*-test for unpaired samples (** p* < 0.05; *** p* < 0.01; **** p* < 0.001).

**Figure 3 cells-12-00608-f003:**
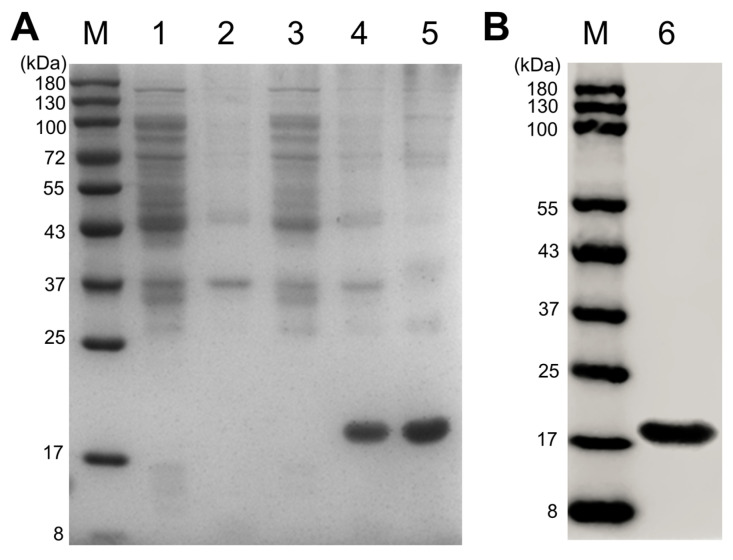
Expression and purification of rTcCTL15. (**A**) Expression of rTcCTL15 from *E. coli* BL21 (DE3) was detected using SDS–PAGE and stained with Coomassie brilliant blue R-250. (**B**) Purification of rTcCTL15 was determined using Western blotting. Molecular weights (kDa) and the positions of standard proteins are marked on the left. Lane M, standard protein marker; lane 1, soluble protein of *E. coli* BL21 (DE3) with pET-28a after induction; lane 2, precipitated protein of *E. coli* BL21 (DE3) with pET-28a after induction; lane 3, soluble protein of *E. coli* BL21 (DE3) with pET-28a-TcCTL15 construct after induction; lane 4, precipitated protein from *E. coli* BL21 (DE3) with pET-28a-TcCTL15 construct after induction; lane 5, highly purified rTcCTL15 protein; lane 6, Western blot analysis of purified rTcCTL15.

**Figure 4 cells-12-00608-f004:**
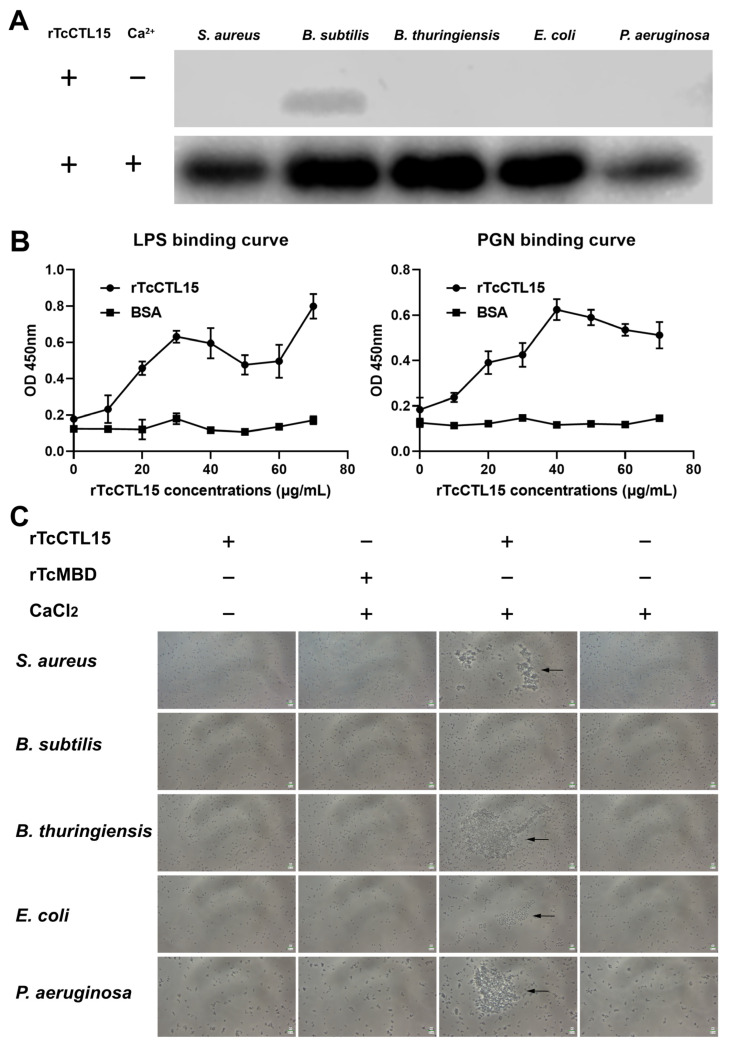
Binding and agglutination activities of rTcCTL15. (**A**) Western blotting was used to detect the binding activities of rTcCTL15 to five microorganisms (the G^+^ bacteria *S. aureus*, *B. subtilis*, and *B. thuringiensis* and the G^-^ bacteria *E. coli* and *P. aeruginosa*). (**B**) ELISA was used to analyze the binding activities of rTcCTL15 to LPS (left) and PGN (right). The results are shown as means ± SDs from three replicates. (**C**) Agglutination activities of rTcCTL15 with either the absence or presence of Ca^2+^. Purified rTcMBD was used as a negative control protein. The arrow points refer to agglutination reactions.

**Figure 5 cells-12-00608-f005:**
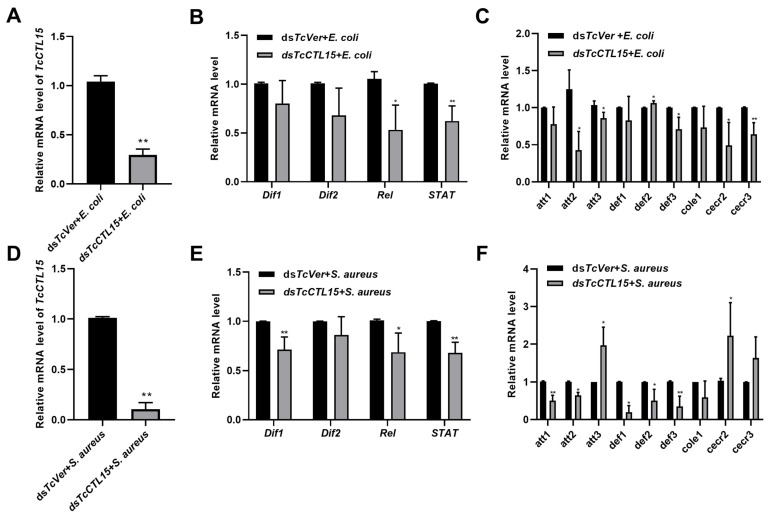
Transcription factor and AMP gene expression upon challenge with *E. coli* or *S. aureus* after successful RNAi-mediated knockdown of *TcCTL15*. (**A**,**D**) RNAi efficiencies after dsRNA injection followed by *E. coli* (**A**) or *S. aureus* (**D**) challenge. (**B**,**E**) The expression profiling of transcription factor genes post-dsRNA-treatment following *E. coli* (**B**) and *S. aureus* (**E**) injections. (**C**,**F**) The expression profiling of AMP genes post-dsRNA-treatment followed by *E. coli* (**C**) and *S. aureus* (**F**) challenge. The bars represent means ± SDs. Each experiment was repeated at least in triplicate. The asterisks denote significant differences from the control groups based on *t*-tests for unpaired samples (** p* < 0.05 and *** p* < 0.01).

**Figure 6 cells-12-00608-f006:**
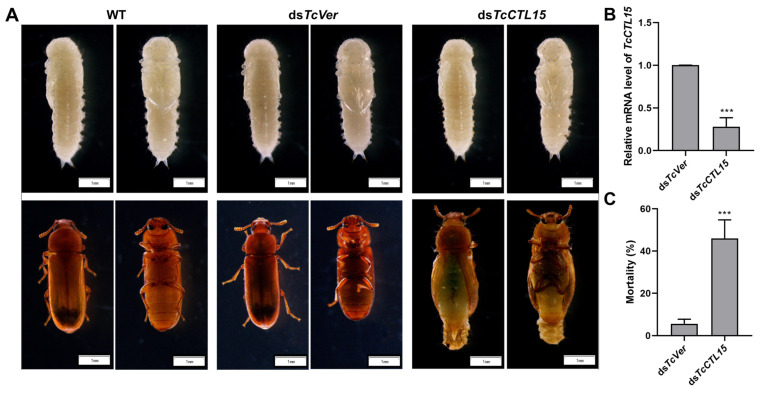
Effect of injecting ds*TcCTL15* in early pupae on transcript levels and metamorphosis. (**A**) Observations of RNAi phenotypes. Scale bar represents 1 mm. (**B**) The RNAi efficiency was detected using qRT-PCR after the injection of dsRNA at 72 h. (**C**) The percent of individuals with abnormal phenotypes and mortality after injecting dsRNA. At least 36 individuals were observed and counted in each group, with three biological replicates. The bars are expressed as means ± SDs. The asterisks denote a significant difference from the control. Differences were calculated using Student’s *t*-test for unpaired samples (**** p* < 0.001).

**Figure 7 cells-12-00608-f007:**
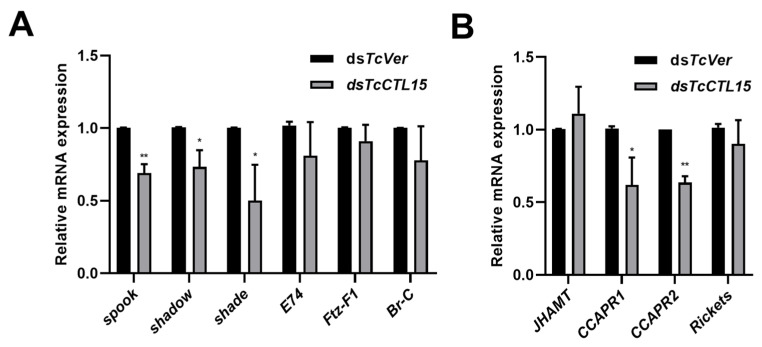
Effect of *TcCTL15* knockdown on the transcript levels of genes involved in the synthesis and metabolism of ecdysone and JH, and eclosion regulation. (**A**) The transcript levels of ecdysone synthesis and metabolism related genes after injecting ds*TcCTL15* in 1-day-old pupae. (**B**) The transcript levels of the key genes of JH biosynthesis and eclosion regulation after injecting ds*TcCTL15* in 1-day-old pupae. The bars are expressed as means ± SDs. The asterisks denote significant differences from the controls. Differences were calculated using Student’s *t*-test for unpaired samples (** p* < 0.05 and *** p* < 0.01).

**Figure 8 cells-12-00608-f008:**
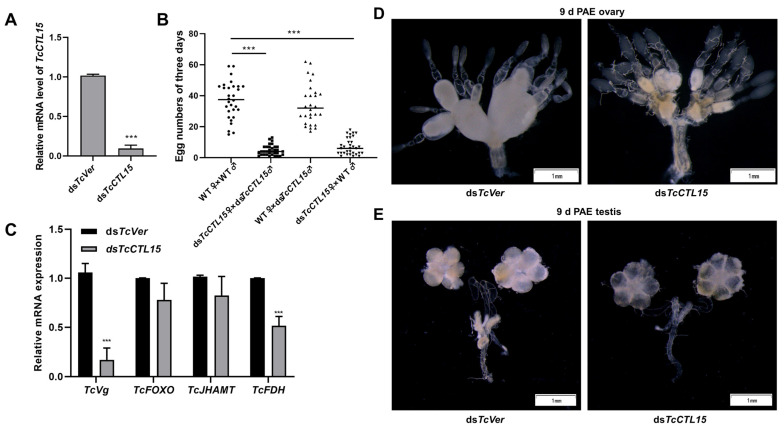
Effect of *TcCTL15* silencing on female fecundity and ovary development. (**A**) The RNAi efficiency of *TcCTL15* after injection of dsRNA at 72 h. (**B**) Effect of *TcCTL15* silencing on female fecundity. (**C**) The relative mRNA levels of reproduction-related genes in ds*TcCTL15*-injected females compared with ds*TcVer*-injected females. The asterisks denote significant differences from the control based on *t*-tests for unpaired samples (**** p* < 0.001). (**D**,**E**) The effect of *TcCTL15* knockdown on ovary (**D**) and testis (**E**) morphology. At least fifteen individuals were dissected and observed for each group. Pupae injected with ds*TcVer* served as a negative control. The scale bar represents 1 mm.

## Data Availability

All data generated or analyzed during this study are included in this article. Other files are attached in the Appendix A.

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
