# Peer review of "Functional Analysis of a Multiple-Domain CTL15 in the Innate Immunity, Eclosion, and Reproduction of Tribolium castaneum"

_cells, 2023, doi:10.3390/cells12040608_

Round 1
Reviewer 1 Report
In this manuscript, Wang et al. identified TcCTL15 , CTL-X of Tribolium castaneum and studied its functions in the innate immunity, eclosion, and reproduction. These findings implicated that TcCTL15 had extensive functions in developmental regulation and antibacterial immunity. Their findings also supported that the TcCTL15 as a candidate target gene for the pest control. The manuscript is written concisely and will be interesting for scientists in the insect field. I recommend acceptation of the manuscript with minor revision.
Majors:
The authors mentioned that knockdown the TcCTL15 affects ecdysone synthesis and eclosion-related regulatory genes. They also found that knockdown the TcCTL15 affects female egg-laying and ovary development. Is TcCTL15 regulate the ovary development via ecdysone synthesis ?
Minors:
line 50: 'D. melanogaster' should be 'Drosophila melanogaster' when used at the first time.
line 124: please use the full name for S. aureus.
line 232: replace 'Drosophila melanogaster' with 'D. melanogaster'
line 246: replace 'upregulated' with 'up-regulated '
Author Response
Dear Dr. Yonggyun Kim and Ms. Maja Arbanas
We greatly appreciate you and the reviewers for thoroughly reviewing our manuscript entitled ‘Functional Analysis of a Multiple Domain CTL15 in the Innate Immunity, Eclosion, and Reproduction of Tribolium castaneum’ and making helpful suggestions to improve it. We found the comments helpful and hope that we have incorporated the suggested improvements to your satisfaction. We considered every comment of the reviewers and addressed the reviewer’s concerns in the revised manuscript. The list of addressed review comments is attached as below. Please e-mail me if you have any questions.
Yours sincerely
Bin Li
Jiangsu Key Laboratory for Biodiversity and Biotechnology, College of Life Sciences, Nanjing Normal University, Nanjing 210023, China;
Reviewer 1# Comments and Suggestions for Authors
In this manuscript, Wang et al. identified TcCTL15, CTL-X of Tribolium castaneum and studied its functions in the innate immunity, eclosion, and reproduction. These findings implicated that TcCTL15 had extensive functions in developmental regulation and antibacterial immunity. Their findings also supported that the TcCTL15 as a candidate target gene for the pest control. The manuscript is written concisely and will be interesting for scientists in the insect field. I recommend acceptation of the manuscript with minor revision.
Majors:
- The authors mentioned that knockdown the TcCTL15 affects ecdysone synthesis and eclosion-related regulatory genes. They also found that knockdown the TcCTL15 affects female egg-laying and ovary development. Is TcCTL15 regulate the ovary development via ecdysone synthesis?
Responses:
Thank you for your questions. The results of Figure 7A showed that the expression of spook, shadow and shade involved in ecdysone (20E) synthesis pathway was down-regulated after injecting dsTcCTL15 in 1-day-old pupae, indicating the eclosion failure caused by TcCTL15 knockdown is through the ecdysone synthesis. The results of Figure 8 revealed that knocking down TcCTL15 after injecting dsRNA in 5-day-old pupae affects female egg-laying and ovary development.
In T. castaneum, the Juvenile hormone is the primary hormone governing reproduction, and ecdysteroids are involved in its oocyte maturation. As shown in Figure 8C, the expression of vitellogenin TcVg and farnesol dehydrogenase TcFDH were significantly down-regulated in dsTcCTL15-injected females, whereas the level of shade, ecdysone receptor (EcR) and ultraspiracle (USP) transcripts exhibited no significant difference between dsTcCTL15 and dsTcVer groups (Figure S1).
Figure S1 The relative mRNA levels of 20E synthesis and metabolism-related genes in dsTcCTL15-injected females compared with dsTcVer-injected females.
The Shade, EcR, and USP genes are involved in the synthesis and metabolism of 20E. Shade, EcR, and USP RNAi severely affected the maturation of the primary oocytes in Tribolium castaneum.
Hence, we concluded that knockdown of TcCTL15 resulted in a reduction in TcVg, and TcFDH at mRNA level, caused female reproductive capacity and ovary development defects and finally led to impaired fecundity, but not TcCTL15 regulate the ovary development via ecdysone synthesis.
References:
Parthasarathy, R.; Sheng, Z.T.; Sun, Z.Y.; Palli, S.R. Ecdysteroid regulation of ovarian growth and oocyte maturation in the red flour beetle, Tribolium castaneum. Insect Biochemistry and Molecular Biology 2010, 40, 429-439, doi: 10.1016/j.ibmb.2010.04.002.
Parthasarathy, R.; Sun, Z.Y.; Bai, H.; Palli, S.R. Juvenile hormone regulation of vitellogenin synthesis in the red flour beetle, Tribolium castaneum. Insect Biochemistry and Molecular Biology 2010, 40, 405-414, doi: 10.1016/j.ibmb.2010.03.006.
Roy, S.; Saha, T.T.; Zou, Z.; Raikhel, A.S. Regulatory Pathways Controlling Female Insect Reproduction. Annu Rev Entomol 2018, 63, 489-511, doi:10.1146/annurev-ento-020117-043258.
Minors:
- line 50: 'D. melanogaster' should be 'Drosophila melanogaster' when used at the first time.
Responses: Thank you for your suggestion. We have replaced D. melanogaster with Drosophila melanogaster Meigen for it was used at the first time, and checked the full name and abbreviation of insect scientific names in the full text.
- line 124: please use the full name for S. aureus.
Responses: We have corrected accordingly in the revised manuscript.
Staphylococcus aureus Rosenbach
- line 232: replace 'Drosophila melanogaster' with 'D. melanogaster'
Responses: We have replaced Drosophila melanogaster with D. melanogaster, and checked the full name and abbreviation of insect scientific names in the full text.
- line 246: replace 'upregulated' with 'up-regulated'
Responses: We have replaced upregualted with up-regulated, and checked in the full text.

Reviewer 2 Report
The paper aims to study the potential multiple roles of transmembrane CTL-Xs 80 in T. castaneum. The paper is well written and organized. The hypothesis of the study is clear and has been achieved. References are adequate, relevant and recent. However, there are minor remarks as follows:
- The major concern is the use of three replicates only in all experiments.
- The abstract has some abbreviations (e.g., PAMPs and CCARPs) that need to be first defined. Although, some of which were defined later in the paper, e.g., PAMPs. The abstract must be self-explanatory.
- Line 31: Replace Toll by Toll receptors or Toll-like receptors.
- Line 45: Write the scientific name in full, with the author of the species (Tribolium castaneum Herbst) when mentioned for the first time. The same observation applies to D. melanogaster (line 50).
- Lines 53 & 54: Delete the sentence stating from “Above all “to the end. This sentence is redundant as the biological functions of CTL-X in insects had already mentioned in lines 50-53 in D. melanogaster with adequate references.
- Line 50: Do not italicize the term “furrowed”.
- Line 69: After inserting the author of the species (Herbst) in line 45, the current authors must delete the author of the species in this line.
- Line 103, central nervous system: Which part of the CNS? Brain, ventral nerve cord or both. Be specific.
- Lines 105 & 106: Specify the age of each stage tested. Early or late larva, pupa….etc is a vague sentence and is not scientific writing.
- Line 124: Write the scientific name and the strain of the bacteria S. aureus and E. coli. The same remark applies to the bacteria in lines 147 & 148. Scientific names with the author of the species/strain must be written in full for the first time. Thereafter, the genus should be abbreviated, and do not write the name of the author of the species again.
- Statistical analysis (lines 213 – 217) (Materials and Methods): The authors stated that one-way ANOVA has been used to analyze the data. This does not apply to all experiments as ANOVA has been applied only for the data shown in Fig. 1. However, in the remaining data (Figs. 2, 5, 6, 7, 8), Student’s t-test has been used. Refer the statistical methods to their respective data. Again, rephrase the last sentence “Values of P<0.05………. in treatments”, as follows:
Statistical significance was set at P ≤ 0.05.
Were the data subjected to normality test before statistical analysis?
-Legend of Fig. 4: The authors stated that the scale bar represents 25 µm. However, scale bar is missing in all photos! In the same Fig., which part is the arrow point? This is missing in the legend of that Fig.
- Legends of Figs. 6 & 7: Which statistical method was used?
- Legend of Fig. 8: Delete the sentence “♀ and ♂ represents females and males, respectively”. ♀ and ♂ are well known biological symbols; thus, no need to explain them.
-Line 376 (Subheading No. 3.6.): Replace ovary development by ovary and testis development.
- Line 395–398: Move the sentence starting from “Taking together …………… impaired fecundity” to the last paragraph of Discussion, as this sentence is good overview of the respective results (i.e., impaired fecundity).
- line 502: Do not italicize “bursicon”, as this is a hormone name rather than a gene name. In the same line, identify first the abbreviation “CCAP”. I think it is (Crustacean cardioactive peptide).
- Line 512: Do insects contain fallopian tubules?!
Author Response
Dear Dr. Yonggyun Kim and Ms. Maja Arbanas
We greatly appreciate you and the reviewers for thoroughly reviewing our manuscript entitled ‘Functional Analysis of a Multiple Domain CTL15 in the Innate Immunity, Eclosion, and Reproduction of Tribolium castaneum’ and making helpful suggestions to improve it. We found the comments helpful and hope that we have incorporated the suggested improvements to your satisfaction. We considered every comment of the reviewers and addressed the reviewer’s concerns in the revised manuscript. The list of addressed review comments is attached as below. Please e-mail me if you have any questions.
Yours sincerely
Bin Li
Jiangsu Key Laboratory for Biodiversity and Biotechnology, College of Life Sciences, Nanjing Normal University, Nanjing 210023, China;
Reviewer 2# Comments and Suggestions for Authors
The paper aims to study the potential multiple roles of transmembrane CTL-Xs 80 in T. castaneum. The paper is well written and organized. The hypothesis of the study is clear and has been achieved. References are adequate, relevant and recent. However, there are minor remarks as follows:
- The major concern is the use of three replicates only in all experiments.
Responses: Thank you for your questions to help improve our paper quality. In our study, all experiments were performed with at least three biological replications. In general, three biological replicates are acceptable for qRT-PCR to test the gene/mRNA expression level.
References:
- Choi, D.; Al Baki, M.A.; Ahmed, S.; Kim, Y. Aspirin Inhibition of Prostaglandin Synthesis Impairs Mosquito Egg Development. Cells 2022, 11, 4092. doi:10.3390/cells11244092
- Huang, Z.; Tian, Z.; Zhao, Y.; Zhu, F.; Liu, W.; Wang, X. MAPK Signaling Pathway Is Essential for Female Reproductive Regulation in the Cabbage Beetle, Colaphellus bowringi. Cells 2022, 11, doi:10.3390/cells11101602.
Besides, for the results of phenotypes of RNAi of TcCTL15, including morphological pictures of pupae and adults, at least 36 individuals in each group with three biological replicates were observed and counted. For the results of morphological pictures of ovary and testis, at least fifteen individuals from each group were dissected and observed. And we have revised this issue accordingly in the legends of figure 6 & 8.
- The abstract has some abbreviations (e.g., PAMPs and CCAPRs) that need to be first defined. Although, some of which were defined later in the paper, e.g., PAMPs. The abstract must be self-explanatory.
Responses: Thank you for your suggestions to improve our publication quality. We have revised the abbreviations accordingly in the revised manuscript.
pathogen-associated molecular patterns (PAMPs)
crustacean cardioactive peptide receptors (CCAPRs)
- Line 31: Replace Toll by Toll receptors or Toll-like receptors.
Responses: We have replaced Toll with Toll-like receptors.
- Line 45: Write the scientific name in full, with the author of the species (Tribolium castaneum Herbst) when mentioned for the first time. The same observation applies to melanogaster (line 50).
Responses: Thank you for your suggestions. We have changed the scientific name in full for the first time and replaced T. castaneum and D. melanogaster with Tribolium castaneum Herbst and Drosophila melanogaster Meigen, respectively.
- Lines 53 & 54: Delete the sentence stating from “Above all “to the end. This sentence is redundant as the biological functions of CTL-X in insects had already mentioned in lines 50-53 in melanogaster with adequate references.
Responses: Thank you for your suggestions. We have deleted this sentence.
- Line 50: Do not italicize the term “furrowed”.
Responses: Changed
- Line 69: After inserting the author of the species (Herbst) in line 45, the current authors must delete the author of the species in this line.
Responses: Changed
- Line 103, central nervous system: Which part of the CNS? Brain, ventral nerve cord or both. Be specific.
Responses: Thank you for your question. The central nervous system of insects includes the brain, thoracic ganglia, and ventral nerve cord. We have revised this sentence as follows:
To manifest tissue distribution, samples were collected from the integument, fat bodies, central nervous system (CNS, including the brain, thoracic ganglia, and ventral nerve cord), gut and hemocytes of 15-day-old larvae.
- Lines 105 & 106: Specify the age of each stage tested. Early or late larva, pupa….etc is a vague sentence and is not scientific writing.
Responses: Thanks for your suggestion. We have revised this sentence to specify the developmental stages, as follows:
early eggs (EE, 1-day-old eggs), late eggs (LE, 3-day-old eggs), early larvae (EL, 1-day-old larvae), late larvae (LL, 20-day-old larvae), early pupae (EP, 1-day-old pupae), late pupae (LP, 5-day-old pupae), early adults (EA, 1-day-old adults), and late adults (LA, 10-day-old adults).
- Line 124: Write the scientific name and the strain of the bacteria aureus and E. coli. The same remark applies to the bacteria in lines 147 & 148. Scientific names with the author of the species/strain must be written in full for the first time. Thereafter, the genus should be abbreviated, and do not write the name of the author of the species again.
Responses: Thank you for your suggestions. We have changed the scientific name of bacteria in full for the first time, and abbreviate the genus name thereafter in the revised manuscript.
Staphylococcus aureus Rosenbach
Escherichia coli Migula
Bacillus subtilis Ehrenberg
Bacillus thuringiensis Berliner
Pseudomonas aeruginosa Schroeter
- Statistical analysis (lines 213 – 217) (Materials and Methods): The authors stated that one-way ANOVA has been used to analyze the data. This does not apply to all experiments as ANOVA has been applied only for the data shown in Fig. 1. However, in the remaining data (Figs. 2, 5, 6, 7, 8), Student’s t-test has been used. Refer the statistical methods to their respective data. Again, rephrase the last sentence “Values of P<0.05………. in treatments”, as follows:
Statistical significance was set at P ≤ 0.05.
Were the data subjected to normality test before statistical analysis?
Responses: Thank you for your questions and suggestions to help improve our publication quality. We have corrected this sentence as follows:
The statistical significance between treatments was assessed using one-way ANOVA followed by Tukey honestly significant difference (HSD) test. Differences between means of treatments and control groups were compared using Student t-test for unpaired samples. Statistical significance was set at P ≤ 0.05.
In addition, all data have been subjected to a normality test before statistical analysis. According to the reviewer’s suggestion, we checked the normality of our data with the Kolmogorov-Smirnov test. If necessary, the statistical method, results, tables and discussion were revised accordingly.
- Legend of Fig. 4: The authors stated that the scale bar represents 25 µm. However, scale bar is missing in all photos! In the same Fig., which part is the arrow point? This is missing in the legend of that Fig.
Responses: Thank you for your questions and suggestions. These pictures have scale bars in Figure 4C. However, they were put together in a set of figures, and the scale bars were too small to recognize. In the revised manuscript, we have replaced this figure with higher-resolution pictures. Also, to make readers easily understand, we have deleted this sentence about the scale bar, because the scale bars were not compulsory to be highlighted in this figure. Moreover, in the same figure legend, we added a description of the arrow point. The arrow point refers to the agglutination reactions.
- Legends of Figs. 6 & 7: Which statistical method was used?
Responses: Thank you for your question. We have revised the figure legends of Fig 6 & 7.
Differences were calculated by Student’s t-test for unpaired samples.
- Legend of Fig. 8: Delete the sentence “♀ and ♂ represents females and males, respectively”. ♀ and ♂ are well known biological symbols; thus, no need to explain them.
Responses: Thank you for your suggestions. We have deleted this sentence.
- Line 376 (Subheading No. 3.6.): Replace ovary development by ovary and testis development.
Responses: Changed
- Line 395–398: Move the sentence starting from “Taking together …………… impaired fecundity” to the last paragraph of Discussion, as this sentence is good overview of the respective results (i.e., impaired fecundity).
Responses: Thank you for your suggestions, we have moved this sentence to the last paragraph of Discussion.
- line 502: Do not italicize “bursicon”, as this is a hormone name rather than a gene name. In the same line, identify first the abbreviation “CCAP”. I think it is (Crustacean cardioactive peptide).
Responses: Thank you for your suggestion. We have changed the “bursicon” and revised the abbreviation “CCAP” as follows:
Insect metamorphosis processes are not only affected by 20E and JH, but also regulated by crustacean cardioactive peptide, bursicon and their receptors.
- Line 512: Do insects contain fallopian tubules?!
Responses: Thank you for your question. Fallopian tubule also called oviduct. In insects, the more common expression was oviduct. So, we replace fallopian tubules by oviducts in the text.
Reference:
Chapman, R. (1998). Reproductive system: Female. In The Insects: Structure and Function (pp. 295-324). Cambridge: Cambridge University Press. doi:10.1017/CBO9780511818202.014
